

# Adaptability and agronomic performance evaluation of mungbean (*Vigna radiata* (L.) Wilczek) varieties under non-inoculated and inoculated rhizobium bacteria conditions

Shiferaw Mebrat Delie[1], Teferi Alem Adamu[1] and Abebaw Mulugeta Andualem[2]

[1] Department of Plant Sciences, College of Agriculture and Environmental Sciences, University of Gondar, Gondar, Amhara, Ethiopia
[2] Department of Horticulture, College of Agriculture and Environmental Sciences, University of Gondar, Gondar, Amhara, Ethiopia

Corresponding author
Shiferaw Mebrat Delie,
smebrat6@gmail.com

## ABSTRACT

**Background**. Because of its high selling price and low cost of production, farmers in the Central Gondar Zone showed great interest in mungbean production. However, the best-adapted varieties and the effectiveness of commercial Rhizobium inoculants were not determined. Hence, a study was conducted for two consecutive years to evaluate the adaptability and agronomic performances of mungbean varieties under non-inoculated and inoculated conditions.

**Methods**. A factorial combination of five mungbean varieties (Shewarobit, NVL-1, Rasa, Boreda, and Arkebe) and two inoculation levels were arranged in randomised complete block design with three replications. Data concerning flowering and maturity, nodulation, plant height, number of main branches and pods, hundred seed weight, biological and seed yields, and harvest index were collected and analyzed using R software.

**Results**. The study provided valuable insights into how variety and inoculation affect mungbean growth and yield. Inoculated seeds of Rasa, Boreda, and Arkebe took longer to flower (42.00, 40.17, and 32.33 days) and mature (86.33, 85.67, and 74.67 days), but in return, they produced more nodules (5.37, 4.83, and 2.77), branches (6.97, 6.75, and 6.45), and pods (11.83, 10.73, and 11.27) than their non-inoculated counterparts. However, no significant differences were seen for Shewarobit and NVL-1 varieties. The 2021 experiment outperformed 2022, showing higher nodules, pods, harvest index, and both biological and seed yields. Variety also played a significant role, influencing stand count, plant height, hundred seed weight, and harvest index. In 2021, the interaction of variety and inoculation affected biological and seed yields, while in 2022, only variety did. Rhizobium inoculation showed inconsistent effects on mungbean yield, suggesting the need for further investigation. The top performers in seed yield were Shewarobit (660.90–1,214.74 kg ha$^{-1}$) and NVL-1 (532.35–1,097.78 kg ha$^{-1}$), followed closely by Rasa (521.99–1,095.00 kg ha$^{-1}$) and Boreda (546.70–1,106.63 kg ha$^{-1}$), with Arkebe (367.85–606.88 kg ha$^{-1}$) yielding the lowest.

**Conclusion**. Mungbean farmers in the Central Gondar Zone should prioritize Shewarobit and NVL-1 for the highest seed yields, with Rasa and Boreda as strong alternatives.

# BACKGROUND AND JUSTIFICATION

## Background

Mungbean (*Vigna radiata* (L.) Wilczek) is a very early maturing and drought-resistant crop with great potential for semi-arid areas (*Georgis & Beshir, 2010*). It is a small herbaceous annual leguminous plant growing to a height of 30 to 120 cm (*Mbeyagala et al., 2017*). The central stems are more or less erect, while the side branches are semi-erect. The leaves are 5–10 cm long trifoliate, with long petioles. Both the stems and leaves are covered with short hairs. The pods are linear, sometimes curved, round, and slender with short pubescence. The seeds are small and nearly globular. The color of the seed is usually green, but yellowish-brown or purple-brown seeds also occur.

Its origin is believed to be in India and Indo-Burmese regions, with the earliest fossilized evidence for mungbean cultivation and consumption found in these areas (*Mbeyagala et al., 2017*). The crop was then spread to most other Asian countries in early times and later to Africa, Australia, and the Americas. Currently, Asia is by far the largest producer of mungbean in the world, contributing 90% of the total production. Mungbean is a recently introduced crop in Ethiopia, where its production is concentrated in drier and marginal areas of Amhara, Benishangul Gumuz, and Southern Nations, Nationalities, and Peoples' Region (SNNPR) (*Ahmed, Siraj & Mohammed, 2017*; *Gereziher et al., 2017*). North Shewa, Oromia special zone, and Southern Wollo are the major mungbean-growing areas in the Amhara region. Due to its drought-tolerant and early-maturing nature, the crop has also been cultivated in East and West Belesa districts since 2017. Because of its high selling price in the market, more than half of the smallholders in the area have started producing the crop.

Mungbean is rapidly gaining importance as a commercial crop in Ethiopia's export market, with production and export volumes significantly increasing (*Habte et al., 2018*). From 2004 to 2013, export volumes grew more than tenfold, with the export value rising from under $2 million to over $27 million USD. As a result of its growing export volume, mungbean was added as the sixth commodity to the Ethiopian Commodity Exchange (ECX) in 2014, marking a significant step in Ethiopia's efforts to diversify its agricultural exports and further contribute to the country's economic development.

## Problem statement and justification

Mungbean productivity under farmer conditions ranged from 0.50 to 1.00 t ha$^{-1}$ (*Kassa et al., 2022*), which is lower than the average productivity of 1.65 t ha$^{-1}$ at the research center and 1.20 t ha$^{-1}$ globally. Ethiopia's low mungbean productivity is caused by a number of limiting factors, such as the unwise use of synthetic fertilizers, scarcity of best-adapted varieties, vulnerability to biotic and abiotic stresses, poor agronomic practices, and the paucity of research on the subject (*Ahmed, Siraj & Mohammed, 2017*; *Gereziher et al., 2017*). Recently, only five mungbean varieties, namely Shewarobit, NVL-1, Arkebe, Rasa,

and Boreda, have been released in Ethiopia. However, the released varieties' agronomic and yield performances have not yet been assessed and evaluated in the Central Gondar Zone.

Thus, the unwise use of synthetic fertilizers is among the major challenges that led to the pollution of air, water and soil. The contaminated soil and water basins, destroys micro-organisms and ecofriendly insects, making the crop more prone to diseases which in turn reduces the soil fertility. Biofertilizers are eco-friendly, productive and economically viable to smaller and average farmers over chemical fertilizers (*Mishra et al., 2013*). Hence, biological nitrogen fixation (biofertilizers) is a key to sustainable mungbean production in tropical soils, which are frequently deficient in nitrogen. Biofertilizers are microorganisms like bacteria, fungi, algal strains playing an important role in improving soil fertility by fixing atmospheric nitrogen and enhancing the quality of nutrient availability in the soil. *Fernandes & Bhalerao (2015)* also reported that mungbean seeds treated with biofertilzers showed significance improvement in the number of leaves, length of leaves, breadth of leaves, length of plant, root length, shoot length and leaf length and total chlorophyll, carbohydrates and protein content contents compared to the untreated counterparts.

However, the Rhizobium bacteria inoculation efficiency largely depends on environmental conditions, types of Rhizobium strains and genotype of the host species (*Giller et al., 2013*; *Franke et al., 2018*). For instance, *Hungria & Vargas (2000)* reported that high temperature and moisture deficiency are major causes of nodulation failure, affecting all stages of the symbiosis and limiting rhizobial growth and survival in soil, including plasmid deletions, genomic rearrangements and reduced diversity. Due to its toxic concentrations of available aluminum and manganese, soil acidity also affects several steps in the development of the symbiosis, including the exchange of molecular signals between the legume and the microsymbiont.

Symbiotic performance of Rhizobium bacteria inoculation is also governed by soil pH, and availability of nutrients such as N, P, and Mo. N, Mo, and P deficiencies, and soil acidity can weaken and limit symbiotic performance (*Polania et al., 2016*; *Haynes & Mokolobate, 2001*; *Asfaw et al., 2012*). For example, *Wolde-meskel et al. (2018)* reported only four out of eight soil types were compatible with chickpea rhizobia inoculants in Ethiopia. *Khan et al. (1999)*, *Bhuiyan et al. (2008)*, and *Rahman et al. (2008)* reported that P and Mo significantly increased the growth, number of nodules, dry matter production, and seed yield of mungbean compared to non-inoculated ones.

The responses of different rhizobial strains also differ on physiological developments and seed yield of mungbean. In line with this, *Sapna (2021)* reported that MR 63 and MB 17a Rhizobia isolate sustained higher yields over the Vigna 703 + Phosphate solubilizing bacteria strain P-36, MR 54, and MH 8b2 Rhizobia strains at both moisture stress and normal conditions. *Musiyiwa, Mpepereki & Giller (2005)* noted that non-native-born Rhizobium species might not always lead to efficient nodulation for mungbean plants because they might be incompatible or present in low population densities. *Hossain & Solaiman (2004)* also showed that Rhizobium strains differ in their nitrogen fixation performances. Additionally, the genetic characteristics of the crop influence inoculation efficiency. *Musiyiwa, Mpepereki & Giller (2005)* demonstrated that certain soybean varieties

were more promiscuous and capable of nodulating with a wider range of Rhizobium strains, suggesting that crop variety plays a role in inoculation success.

### Research need and rationale

Despite these findings, the Menagesha Bioteck industry in Ethiopia produces Rhizobium inoculants without considering the environmental and genetic factors for the host species or bacterial strains. No research has been done to evaluate the performance of commercially produced Rhizobium inoculants in the Central Gondar Zone, despite their potential for sustainable agriculture. Therefore, it is essential to assess the performance of these inoculants across different growing seasons, environments, and mungbean genotypes. Comparing results from different seasons helps evaluate how environmental factors, such as temperature, rainfall, and soil conditions, affect mungbean performance. This comparison also ensures the consistency of the findings, making sure trends are not influenced by seasonal variations. In light of these considerations, the present study aims to evaluate the response of five mungbean varieties to two levels of Rhizobium inoculation, focusing on nodulation, plant height, number of main branches, number of pods, hundred seed weight, seed and biological yields, and harvest index.

## METHODOLOGY

### Description of the study area

The study was conducted for two consecutive years (2021 and 2022) during the main cropping season, spinning the months from July to November at the University of Gondar Experimental and Demonstration Site in Arbya, West Belesa district of Ethiopia (Fig. 1). It is a drought-prone and food-insecure province located at 120 14′60″N latitude and 37044′60″E longitude. It has also altitudinal ranges of 1,777 to 1,806 masl with an annual rainfall amount of 800 to 1,200 mm. According to *Wubie (2019)*, the temperature ranges from 13.84–28.04 °C to 13.07–26.75 °C. Black vertisol soil predominates at the experimental site, which is located at an elevation of 1,680 m above sea level. Agriculture is the dominant income source for the communities residing in the area. The West Belesa farmers are mainly growing sorghum, chickpea, teff, and currently mungbean in the crop production sector and rare cattle, donkey, goat, and poultry in the livestock sector. Mena, Hota, and Balagas are the unutilized year-round rivers found in the district.

### Description of varieties utilized in the experiment

The study utilized five mungbean varieties: Shewarobit, NLV-1, Arkebe (SML-668), Rasa (N-26), and Boreda (MH-97-6). Shewarobit variety is in some cases considered a local cultivar for the reason that this variety was introduced in Ethiopia during the starting time of mungbean cultivation. Grain collectors and buyers were paying a relatively higher price for Rasa, Boreda, NVL-1, and Arkebe varieties compared to the Shewarobit variety for the larger seed size and deep green color of the former varieties than the latter ones. The certified seeds of these varieties were sourced and collected from Humera, Melkassa, Debre Birhan, and Hawassa Agricultural Research Centres. Table 1 provides detailed descriptions of the varieties.

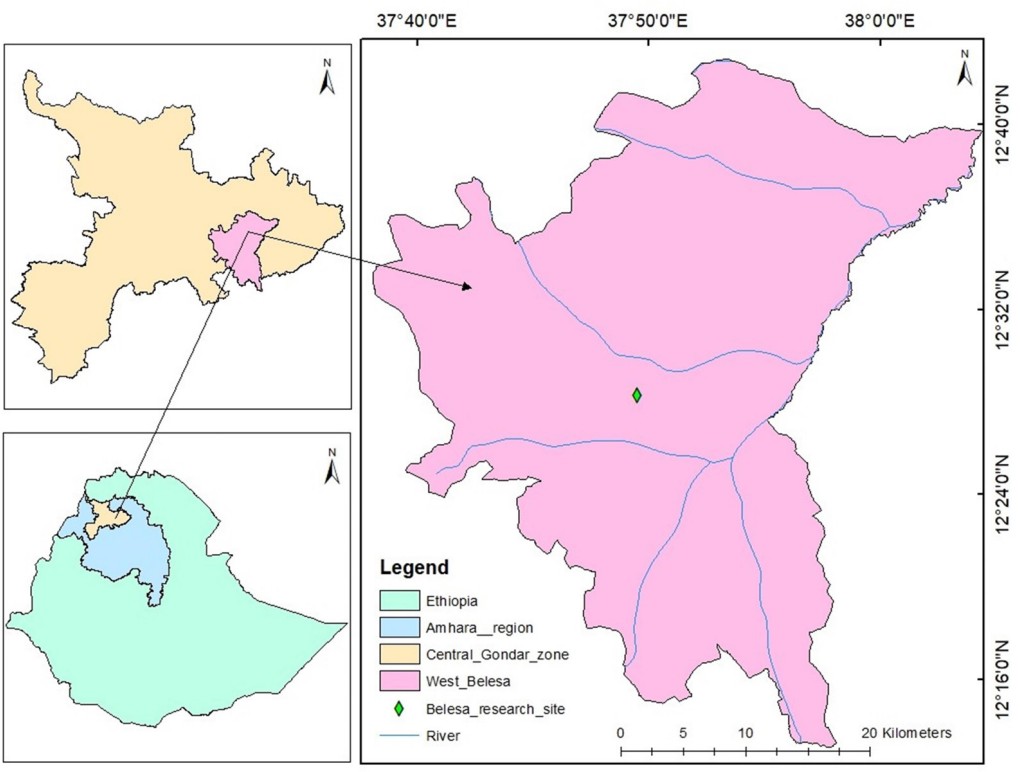

**Figure 1  Geographical map of the study area.**

**Table 1  Description of released mungbean varieties used in the experiment.**

| Characteristics | Varieties | | | | |
|---|---|---|---|---|---|
| | Shewarobit | Rasa | NVL-1 | Boreda | Arkebe |
| Year released/introduced | 2011 | 2011 | 2014 | – | 2014 |
| Research centre | – | Melkasa | Melkasa | Hawasa | Humera |
| Altitude (masl) | 900–1,670 | 900–1,670 | 450–1,650 | 550–1,780 | 600–1,000 |
| Rainfall (mm) | 350–550 | 350–550 | 300–750 | – | 400–800 |
| Plant height (cm) | – | 33 | 40–50 | – | 38–58 |
| Growth habit | – | Determinate bush | Determinate | – | Erect |
| Seed colour | – | Green | Shiny green | – | Green |
| Days to maturity | 75–90 | 65–80 | 60–70 | 70–90 | 60–68 |
| Disease reaction | – | Resistant | Resistant | – | – |
| Yield on research (kg ha$^{-1}$) | 800–1,500 | 800–1,500 | 750–1,500 | 650–1,000 | 1,955–2,526 |
| Yield on farmer (kg ha$^{-1}$) | 500–1,000 | 500–1,000 | 500–1,500 | – | – |

**Notes.**
Source: *MOA, 2011*; *MOA, 2014*; *Kassa et al., 2018*.

## Source of rhizobia inoculant and method of seed inoculation

Commercially produced mungbean rhizobium bacteria inoculant was collected from the Menagesha Biotechnology Industry. Seed inoculation was made before sowing using the

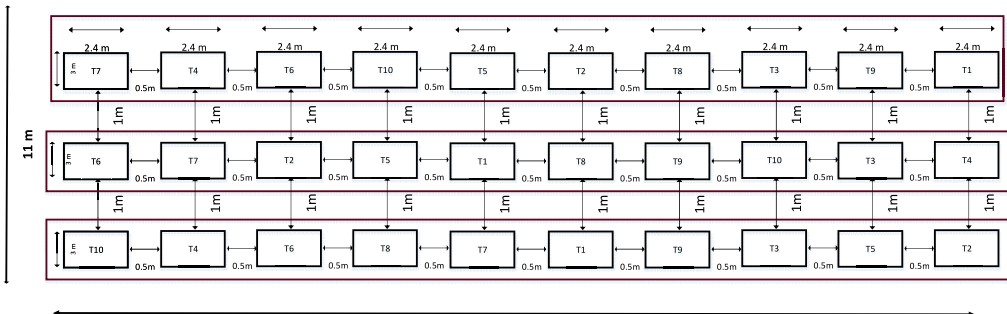

**Treatment descriptions**

T1 = Rhizobium bacteria un-inoculated NVL-1 variety;
T2 = Rhizobium bacteria un-inoculated Arkebe variety;
T3 = Rhizobium bacteria un-inoculated Shewa robit variety;
T4 = Rhizobium bacteria un-inoculated Rasa (N-26) variety;
T5 = Rhizobium bacteria un-inoculated Boreda variety;

T6 = Rhizobium bacteria inoculated NLV-1 variety
T7 = Rhizobium bacteria inoculated Arkebe variety
T8 = Rhizobium bacteria inoculated Shewarobit variety
T9 = Rhizobium bacteria inoculated Rasa (N-26) variety
T10 = Rhizobium bacteria inoculated Boreda variety

**Figure 2** Layout of the experiment.

procedure developed by *Ethiopian Institute of Agricultural Research (2003)*. To guarantee that the applied inoculant adhered to the seeds, the sticker was prepared by adding and thoroughly mixing two tablespoons of table sugar to the 300 ml of clean water. Then, the sticker was added and evenly mixed to clean seed lot sufficient to 0.25 hectare. The excess sticker solution was properly drained from the seed before adding biofertilizer. After being air dried under shade, the inoculated seeds were then sown into their respective plots at the recommended spacing. To prevent contamination, plots with non-inoculated seeds were planted first, followed by the inoculated counterparts.

## Treatments and experimental design

The experiment contained five mungbean varieties (Table 1) and two rhizobium bacteria inoculation levels, *i.e.,* inoculated and non-inoculated ones. A 5 × 2 factorial randomized complete block design (RCBD) with three replications (Fig. 2) was utilized. The gross plot size was 2.4 m × 3 m (7.2 m$^2$). Spacing of 0.5 m between plots and one m between blocks were allocated. Each gross plot was accommodated by six rows, from which the middle four rows were harvested for the data source. Hence, the net plot size of this experiment was 1.8 m × 3 m (5.4 m$^2$).

## Management of the experiment

All agronomic activities were implemented in harmony with farmers' practices, except the treatments. The land was ploughed two times by oxen, and leveling was done manually before sowing. Sowing was carried out in the second and third weeks of July for two consecutive years (2021 and 2022, respectively) by placing two seeds per hole and thinning them to one plant after full emergence. The one weak difference in sowing dates between 2021 and 2022 was due to the delayed onset of the rainy season in 2022. As per the practice of farmers in the West Belesa district, no fertilizer was applied. Hand weeding was

implemented 3, 5, and 7 weeks after sowing. Endosulfan with 35% emulsifier concentrate and 472 g active ingredient per hectare was utilized for the management of pod borer just after pod development. Harvesting was performed manually, pulling the plant from the soil, in the first and second weeks of November. Then, threshing was done by beating with a stick. The principle of blocking in RCBD was respected during data recording, weeding, harvesting, and threshing operations.

## Data collection and measurement

Days to 50% flowering were determined by counting the number of days taken from sowing to the duration when 50% of the plants in a plot settled first flower. Days to 90% physiological maturity were taken by counting the number of days taken from sowing to the duration when 90% of the plants in a plot disclosed a canopy color change to dark brown and dried. The number of nodules per plant was determined by counting its number from the average of five randomly selected plants at the mid-flowering period. Fresh nodule weight was recorded immediately after collection using a sensitive balance. Plant height was measured from the surface of the ground to the tip of the main stem of the plant using a small meter indiscriminately taken of the average five plants in a plot during physiological maturity. The number of primary branches per plant was determined by counting the number of branches grown on the main stem of five arbitrarily taken plants in a plot at maturity. A stand count was recorded by counting the number of survived plants at physiological maturity from a net plot area.

The number of pods per plant was recorded by counting the total number of pods of five randomly taken plants from the net plot area at harvesting time. The biological yield was determined by drying and uprooting the above-ground biomass from the whole net plot area in the open air (including the seed yield) and weighing using a spring balance. Hundred seed weights were determined by weighing 100 randomly taken dry seeds using a sensitive balance, and the weight was adjusted to 10% moisture level. Seed yield was measured from the air-dried seeds of the net plot area using a sensitive balance and adjusted at 10% seed moisture content. Harvest index was computed as the ratio of seed yield to biomass yield.

## Statistical data analysis

The collected raw data were subjected to analysis of variance (ANOVA) according to the generalized linear model (GLM) procedure of R version 4.2 (*R Core Team, 2022*), and interpretation of the results was made following the procedure of *Gomez & Gomez (1984)*. Homogeneity of variance among years for the collected parameters was conducted through Levene's test. Furthermore, treatment mean separations were done using the least significance difference (LSD) test at a 5% level of significance.

## RESULTS

### Levene's test results for homogeneity of variances among years

Levene's test for homogeneity of variances between 2021-year and 2022-year experiments revealed no significant differences for days to 50% flowering (DF), days to 90% physiological maturity (DM), number of nodules per plant (NNPP), fresh weight of the nodules per

**Table 2** Days to 50% flowering (DF), days to 90% physiological maturity (DM), plant height (PH), number of primary branches per plant (NPBPP), stand count (SC), hundred seed weight (HSW) and harvest index (HI) of mungbean as affected by the main effect of sowing year.

| Years | DF (days) | DM (days) | PH (cm) | NPBPP (no) | SC (no) | HSW (g) | HI (%) |
|---|---|---|---|---|---|---|---|
| **2021** | 40.80a | 85.00a | 30.24a | 6.83a | 98.33 | 4.43 | 29.40a |
| **2022** | 38.83b | 83.13b | 28.61b | 6.09b | 98.03 | 4.46 | 22.26b |
| **LSD$_{0.05}$** | 0.53 | 0.56 | 1.56 | 0.20 | ns | ns | 1.09 |
| **CV (%)** | 2.52 | 1.28 | 10.13 | 5.84 | 7.73 | 4.59 | 8.02 |

Notes.
Means followed by the same letter(s) in columns are not significantly different at 5% level by the LSD test.

plant (FNPP), stand count (SC), plant height (PH), number of primary branches per plant (NPBPP), number of pods per plant (NPPP), hundred seed weight (HSW), and harvest index (HI) (Table S1). However, very highly significant differences were observed in biological (BY) and seed yield (SY). Consequently, combined mean square ANOVA values were used for DF, DM, SC, PH, NPBPP, NPPP, HSW, and HI (Table S2). In contrast, separate mean square ANOVA values were utilized for the biological and seed yield in 2021 and 2022 (Table S3).

## Effects of sowing year on phenological, growth, yield components and yield

The main effect of the year showed a very highly significant effect for days to 50% flowering, days to 90% physiological maturity, number of main branches per plant, number of pods per plant, plant height ($p < 0.05$), and harvest index (Table S2). Conversely, the two-way (year by varieties and year by inoculation) and three-way interaction (year by varieties by inoculation) effects revealed non-significant effects on DF, DM, PH, NPBPP, and HI. Additionally, stand count and hundred seed weight were not affected by the main effects of year and its interaction effects. The non-significant response of SC and HSW across years indicated that these parameters responded similarly, regardless of the year.

The 2021-year experiment recorded longer days to 50% flowering (40.63 days), 90% maturity (84.80 days), taller plant height (30.24 cm), more primary branches plant$^{-1}$ (6.80), higher number pods plant$^{-1}$ (12.92), and a greater harvest index (29.31%) compared to the 2022-year experiment, which had lower DF (38.60 days), DM (82.83 days), shorter PH (28.61 cm), lower NPBPP (6.03), NPPP (10.78), and HI (22.04%) (Table 2).

## Effects of varieties and rhizobium inoculation on crop phenology and growth
### Days to 50% flowering and 90% physiological maturity
The main effects of varieties, inoculation, and year ($p < 0.01$) along with the interaction effects of varieties by inoculation ($p < 0.05$) demonstrated a significant difference for days to 50% flowering (Table S2). Both the inoculated and non-inoculated Shewarobit variety required the longest days to 50% flowering (47.00 days), followed by the inoculated NVL-1 variety, which took 41.67 days (Table 3). In contrast, the non-inoculated Arkebe variety (30.00 days) took the shortest days to reach 50% flowering.

**Table 3** Days to 50% flowering and 90% physiological maturity of mungbean as affected by the interaction effects of varieties and Rhizobium bacteria inoculation levels.

| Varieties | Days to 50% flowering (days) | | Days to 90% physiological maturity (days) | |
|---|---|---|---|---|
| | Un-inoculated | Inoculated | Un-inoculated | Inoculated |
| Shewarobit | 47.00a | 47.00a | 94.00a | 93.50a |
| NVL-1 | 40.00c | 41.67b | 82.50c | 85.83b |
| Rasa | 40.00c | 42.00b | 82.67c | 86.33b |
| Boreda | 38.00d | 40.17c | 83.17c | 85.67b |
| Arkebe | 30.00f | 32. 33e | 72.33e | 74.67d |
| LSD$_{0.05}$ | 1.18 | | 1.26 | |
| CV (%) | 2.52 | | 1.28 | |

Notes.

Means followed by the same letter(s) in columns and rows are not significantly different at 5% level by the LSD test.

**Table 4** Number of nodules per plant, fresh weight of nodules per plant and number of pods per plant of mungbean as affected by the interaction effects of varieties and years.

| Varieties | Number of nodules per plant (number) | | Fresh weight of nodules per plant (mg) | | Number of pods per plant (number) | |
|---|---|---|---|---|---|---|
| | 2021 | 2022 | 2021 | 2022 | 2021 | 2022 |
| Shewarobit | 8.57a | 7.43b | 0.61a | 0.54b | 17.50a | 13.13b |
| NVL-1 | 7.40b | 4.90d | 0.54b | 0.36cd | 13.27b | 11.47cd |
| Rasa | 5.77c | 4.50e | 0.40c | 0.32d | 12.23bc | 9.73ef |
| Boreda | 5.00d | 3.97f | 0.35d | 0.27e | 10.70de | 9.03f |
| Arkebe | 3.03g | 2.30h | 0.25e | 0.18f | 12.33bc | 10.53de |
| LSD$_{0.05}$ | 0.27 | | 0.05 | | 1.34 | |
| CV (%) | 4.40 | | 11.00 | | 9.51 | |

Notes.

Means followed by the same letter(s) in columns and rows are not significantly different at 5% level by the LSD test.

Similarly, days to 90% physiological maturity were very highly influenced by the main effects of variety, inoculation, and the interaction effects of varieties and inoculation (Table S2). The Shewarobit variety, whether inoculated (93.50 days) or non-inoculated (94.00 days) conditions, took the longest days to reach 90% physiological maturity, followed by inoculated Rasa (86.33 days), NVL-1 (85.83 days), and Boreda (85.67 days) varieties (Table 3). While the shortest days to 90% physiological maturity were recorded for the non-inoculated Arkebe variety.

### Number and fresh–weight of nodules per plant

The main and interaction effects of varieties and year showed highly significant differences in the number of nodules plant$^{-1}$ (Table S2). Variety Shewarobit produced the highest number of nodules plant$^{-1}$ (8.57) in the 2021-year experiment, followed by the same variety in the 2022-year experiment (7.43), and variety NVL-1 (7.40) in the 2021 experiment (Table 4). The lowest number of nodules in plant$^{-1}$ was observed from Arkebe varieties during the 2022-year experiment.

Additionally, the main and interaction effects of inoculation and growing years showed significant differences for the number of nodules plant$^{-1}$ (Table S2). In the 2021-year

**Table 5 Number of nodules per plant and fresh weight of nodules per plant of mungbean as affected by the interaction effects of Rhizobium bacteria inoculation levels and growing years.**

| Rhizobium bacteria inoculation levels | Number of nodule per plant (number) | | Fresh weight of nodules per plant (mg) | |
|---|---|---|---|---|
| | 2021 | 2022 | 2021 | 2022 |
| Un-inoculated conditions | 5.69b | 4.56c | 0.40b | 0.33c |
| Inoculated conditions | 6.21a | 4.68c | 0.46a | 0.34c |
| LSD$_{0.05}$ | 0.17 | | 0.03 | |
| CV (%) | 4.40 | | 11.00 | |

Notes.
Means followed by the same letter(s) in columns and rows are not significantly different at 5% level by the LSD test.

**Table 6 Number of nodules, primary branches and pods per plant as affected by the interaction effects of varieties and Rhizobium inoculation.**

| Varieties | Number of nodules per plant (number) | | Number of primary branches per plant (number) | | Number of pods per plant (number) | |
|---|---|---|---|---|---|---|
| | Un-inoculated | Inoculated | Un-inoculated | Inoculated | Un-inoculated | Inoculated |
| Shewarobit | 8.03a | 7.97a | 7.07a | 6.98a | 15.87a | 14.77a |
| NVL-1 | 6.00c | 6.30b | 6.50bc | 6.93ab | 12.43b | 12.30b |
| Rasa | 4.90e | 5.37d | 6.00d | 6.97a | 10.13de | 11.83bc |
| Boreda | 4.13f | 4.83e | 5.50e | 6.75abc | 9.00e | 10.73cd |
| Arkebe | 2.57g | 2.77g | 5.45e | 6.45c | 11.83bc | 11.27bcd |
| LSD$_{0.05}$ | 0.27 | | 0.44 | | 1.34 | |
| CV (%) | 4.40 | | 5.84 | | 9.51 | |

Notes.
Means followed by the same letter(s) in columns and rows are not significantly different at 5% level by the LSD test.

experiment, the inoculated seeds produced the highest number of nodules plant$^{-1}$ (6.21), followed by the same year experiment in the non-inoculated conditions (5.69) (Table 5). In contrast, both the inoculated (4.68) and non-inoculated (4.56) seeds in the 2022-year experiment provided the lowest numbers of nodules plant$^{-1}$.

The number of nodules plant$^{-1}$ was also highly influenced by the main and interaction effects of varieties and Rhizobium bacteria inoculation levels (Table S2). The Shewarobit variety brought the highest number of nodules plant$^{-1}$ at both inoculated (8.03) and non-inoculated (7.97) conditions (Table 6). The Arkebe variety at non-inoculated conditions yielded the lowest number of nodules plant$^{-1}$.

Moreover, the main effects of years, varieties, and inoculation ($p < 0.01$) and the interaction of effects of year by variety and year by inoculation ($p < 0.05$) showed significant differences on the fresh weight of nodules plant$^{-1}$ (Table S2). However, the other two-way (variety by inoculation) and three-way (year by variety by inoculation) interaction effects were non-significant. The Shewarobit variety in the 2021-year experiment produced the highest fresh weight of nodules plant$^{-1}$ (0.61 mg), while the Arkebe variety recorded the lowest in the 2022-year experiment (Table 4). Moreover, in 2021, inoculated mungbean seeds produced significantly higher fresh weight of nodules plant$^{-1}$ (0.46 mg) (Table 5) than their non-inoculated counterparts (0.33 mg).

**Table 7  Stand count, plant height, hundred seed weight and harvest index of mungbean as affected by the main effects of varieties and Rhizobium bacteria inoculation levels.**

| Varieties | Stand count (no) | Plant height (cm) | Hundred seed weight (g) | Harvest index (%) |
|---|---|---|---|---|
| Shewarobit | 107.75ab | 30.93a | 3.52c | 28.444a |
| NVL-1 | 109.83a | 30.52a | 4.91a | 24.264b |
| Rasa | 103.92ab | 29.33ab | 4.97a | 27.444a |
| Boreda | 101.50b | 29.02ab | 5.06a | 27.70a |
| Arkebe | 67.92c | 27.32b | 3.78b | 21.31c |
| LSD$_{0.05}$ | 6.28 | 2.46 | 0.17 | 1.72 |
| **Rhizobium bacteria inoculation levels** | | | | |
| Un-inoculated | 98.80 | 29.76 | 4.42 | 25.99 |
| Inoculated | 97.57 | 29.09 | 4.47 | 25.67 |
| LSD$_{0.05}$ | ns | ns | ns | ns |
| CV (%) | 7.73 | 10.13 | 4.59 | 8.02 |

Notes.
   Means followed by the same letter(s) in a column are not significantly different at 5% level by the LSD test.

## Stand count

The analysis of variance showed that only the main effects of varieties showed very highly significant differences in stand count at maturity (Table S2). However, the main effects of inoculation as well as all interaction effects were non-significant on stand count at maturity. The NVL-1 variety recorded the highest stand count at physiological maturity with 109.75 plants (representing 91.46% survival), followed by Shewarobit with 107.75 plants (89.79% survival), Rasa with 103.92 plants (86.66% survival), and Boreda varieties with 101.58 plants (84.58% survival) (Table 7). In contrast, the lowest stand count with 67.92 plants (56.60% survival) was observed in the Arkebe variety.

## Plant height

The main effects of varieties showed significant differences in plant height (Table S2). However, the main effect of Rhizobium inoculation and all the two (year by varieties, year by inoculation, and varieties by inoculation) and three-way (year by varieties by inoculation) interaction effects were non-significant for the plant height. The tallest plant height (30.93 cm) was recorded for the Shewarobit variety, while the shortest height (27.32 cm) was observed for the Arkebe variety (Table 7).

## Number of primary branches per plant

The main and interaction effects of varieties and inoculation showed very highly significant differences in the number of primary branches plant$^{-1}$ (Table S2). The Shewarobit non-inoculated (7.07) variety had the highest number of primary branches plant$^{-1}$ while the Arkebe non-inoculated variety had the lowest (Table 6). Significant varietal differences on number of primary branches plant$^{-1}$ were found both in inoculated and non-inoculated conditions.

**Table 8** Biological and seed yield (kg ha$^{-1}$) of mungbean as affected by the interaction effects of varieties and Rhizobium bacteria inoculation levels in 2021.

| Varieties | Biological yield (kg ha$^{-1}$) | | Seed yield (kg ha$^{-1}$) | |
|---|---|---|---|---|
| | Un-inoculated | Inoculated | Un-inoculated | Inoculated |
| Shewarobit | 3,750.00abc | 3,486.11bc | 1,214.74a | 1,107.06a |
| NVL-1 | 3,819.44ab | 3,995.83a | 1,097.78a | 1,063.68a |
| Rasa | 2,883.89d | 3,336.11c | 885.83b | 1,095.00a |
| Boreda | 2,774.00de | 3,466.67bc | 850.71b | 1,106.63a |
| Arkebe | 1,944.44f | 2,430.56e | 459.51c | 606.88c |
| LSD$_{0.05}$ | 438.17 | | 162.14 | |
| CV (%) | 8.01 | | 9.96 | |

**Notes.**
Means followed by the same letter(s) in columns and rows are not significantly different at 5% level by the LSD test.

## Effects of varieties and rhizobium inoculation on yield components and yield

### Number of pods per plant

The ANOVA results showed that the main ($p < 0.01$) and interaction ($p < 0.05$) effects of varieties and years showed significant differences in the number of pods plant$^{-1}$ (Table S2). The highest number of pods plant$^{-1}$ was recorded for the Shewarobit variety in the 2021-year experiment (17.50), followed by the same variety in the 2022-year experiment (13.13) (Table 4). However, the lowest number of pods plant$^{-1}$ was observed for the Boreda variety in the 2022-year experiment.

Moreover, the interaction effects of varieties and inoculation levels showed a significant difference in the number of pods plant$^{-1}$ (Table S2). However, all other interaction effects (year by inoculation and year by varieties by inoculation) and the main effect of inoculation were non-significant. The highest number of pods plant$^{-1}$ (15.87) was obtained for the non-inoculated Shewarobit variety and the lowest (11.27) for the inoculated Arkebe variety (Table 6).

### Biological yield

The analysis of variance revealed that the biological yield of mungbean was significantly affected by the main and interaction effects of varieties and inoculation in the 2021 experiment (Table S3). The highest biological yield (3,995.83 kg ha$^{-1}$) was recorded for the inoculated NVL-1, followed by the non-inoculated NVL-1 (3,819.44 kg ha$^{-1}$) and Shewarobit (3,750.00 kg ha$^{-1}$) varieties (Table 8). The lowest biological yield (1,944.44 kg ha$^{-1}$) was obtained from the non-inoculated Arkebe variety.

However, the 2022-year experiment analysis of variance indicated that only the main effect of varieties showed significant differences in the biological yield of mungbean (Table S3). All other effects- (the main effects of inoculation as well as the interaction effects of year by varieties, year by inoculation, varieties by inoculation, and year by varieties by inoculation) were non-significant on biological yield. The highest biological yield was recorded for the Shewarobit (2,662.50 kg ha$^{-1}$) and NVL-1 (2,561.11 kg ha$^{-1}$) varieties, followed by Rasa (2,255.56 kg ha$^{-1}$) and Boreda (2,252.78 kg ha$^{-1}$) varieties

Table 9 Biological and seed yield of mungbean as affected by the main effects of varieties and Rhizobium bacteria inoculation levels in 2022.

| Varieties | Biological yield (kg ha$^{-1}$) | Seed yield (kg ha$^{-1}$) |
|---|---|---|
| Shewarobit | 2,662.50a | 660.90a |
| NVL-1 | 2,561.11a | 532.35b |
| Rasa | 2,255.56b | 521.99b |
| Boreda | 2,252.78b | 546.70b |
| Arkebe | 2,000.00c | 367.85c |
| LSD$_{0.05}$ | 250.42 | 100.75 |
| **Rhizobium bacteria inoculation levels** | | |
| Un-inoculated | 2,306.67 | 530.83 |
| Inoculated | 2,386.11 | 521.07 |
| LSD$_{0.05}$ | ns | ns |
| CV (%) | 8.80 | 15.79 |

Notes.
Any treatment means not sharing a letter in common differs statistically at the 5% probability level; ns, non-significant.

(Table 9). While the lowest biological yield (2,000.00 kg ha$^{-1}$) was recorded for the Arkebe variety.

### Hundred seed weight

Only the main effects of varieties showed very significant effects on hundred seed weight (Table S2). However, the main effects of Rhizobium inoculation levels and year, as well as all interaction effects, were non-significant on hundred seed weight. The highest hundred seed weight was observed for Boreda (5.06 g), Rasa (4.97 g), and NVL-1 (4.91 g) varieties, followed by the Arkebe variety (3.78 g) (Table 7). In contrast, the lowest hundred seed weight was recorded for the Shewarobit variety (3.52 g).

### Seed yield

The analysis of variance for the 2021-year experiment showed that the main and interaction effects of varieties and inoculation had significant influences on seed yield of mungbean (Table S3). The highest seed yield (1,214.74 kg ha$^{-1}$) was recorded for the non-inoculated Shewarobit variety, and the lowest seed yield (459.51 kg ha$^{-1}$) was recorded for the non-inoculated Arkebe variety (Table 8).

Nevertheless, the 2022-year experiment analysis of variance indicated that only the main effects of varieties had a very highly significant effect on seed yield (Table S3). All other effects, *i.e.*, the main effect of inoculation and the interaction effect of variety and inoculation, were non-significant. The highest seed yield (660.90 kg ha$^{-1}$) was recorded for the Shewarobit variety, followed by the NVL (532.35 kg ha$^{-1}$), Rasa (521.99 kg ha$^{-1}$), and Boreda (546.70 kg ha$^{-1}$) varieties (Table 9). The lowest seed yield (367.85 kg ha$^{-1}$) was observed for the Arkebe variety.

### Harvest index

Only the main effects of varieties showed very high significant differences in harvest index (Table S2). However, all the two-way (year by varieties, year by inoculation, and varieties by inoculation) and three-way (year by varieties by inoculation) interactions and the

main effects of inoculation revealed non-significant effects on harvest index. The highest harvest index was recorded for Shewarobit (28.21%), Rasa (27.04%), and Boreda (28.10%) varieties, followed by the NVL-1 variety (24.26%) (Table 7). The lowest harvest index (20.76%) was obtained from the Arkebe variety.

### Correlation analyses

The correlation analyses revealed highly significant positive associations between the days to flowering and maturity with several parameters: number of nodules per plant ($r = 0.93$** and $0.90$**, respectively), fresh weight of nodules per plant ($r = 0.87$** and $0.85$**), stand count ($r = 0.75$** and $0.71$**), plant height ($r = 0.39$** and $0.37$**), number of primary branches per plant ($r = 0.64$** and $0.59$**), number of pods per plant ($r = 0.55$** and $0.54$**), and harvest index ($r = 0.58$** and $0.56$**) (Fig. 3). Additionally, the number of nodules per plant and the fresh weight of nodules per plant exhibited strong positive correlations with stand count ($r = 0.69$** and $0.64$**), plant height ($r = 0.47$** and $0.49$**), number of primary branches per plant ($r = 0.70$** and $0.68$**), number of pods per plant ($r = 0.70$** and $0.71$**), and harvest index ($r = 0.61$** and $0.57$**). Furthermore, the number of pods per plant showed a highly significant positive association with the number of primary branches per plant ($r = 0.68$**). However, the weight of one hundred seeds had a significant negative correlation with the number of pods per plant ($r = -0.51$**) and a positive correlation with stand count ($r = 0.35$**), but showed no associations with other parameters.

Similarly, both biological yield and seed yield were found to significantly correlate with days to flowering ($r = 0.66$** for both) and maturity ($r = 0.61$** and $0.62$**). Strong positive associations were also observed with the number of nodules per plant ($r = 0.78$** and $0.76$**), fresh weight of nodules per plant ($r = 0.76$** and $0.72$**), stand count ($r = 0.59$** and $0.55$**), plant height ($r = 0.46$** and $0.51$**), number of primary branches per plant ($r = 0.75$** and $0.74$**), and number of pods per plant ($r = 0.63$** and $0.65$**). Notably, seed yield displayed strong positive associations with biological yield ($r = 0.94$**) and harvest index ($r = 0.89$**). In conclusion, the correlation analyses indicate that seed yield in mungbean is largely influenced by the number of days to flowering and maturity, alongside the nodule, branch, and pod development characteristics of the genotypes.

## DISCUSSION

### Implications of sowing year on phenological, growth and yield components

The 2021-year experiment took 4.83% longer days to flowering, 2.20% longer days to maturity, 5.39% taller plant height, 10.83% higher number of primary braches plant$^{-1}$, and 24.29% larger harvest index than the 2022-year experiment. The shorter days to flowering, maturity, and plant height, together with lower number of primary branches plant$^{-1}$, number of pods plant$^{-1}$, and harvest index in the 2022-year experiment, might be due to the high temperature and moisture deficiency of the second year (2022) compared to the 2021-year. This result is in agreement with *Dinsa et al. (2022)*, who reported that year had significant influence on number of pods plant$^{-1}$, number of seed pod$^{-1}$, and

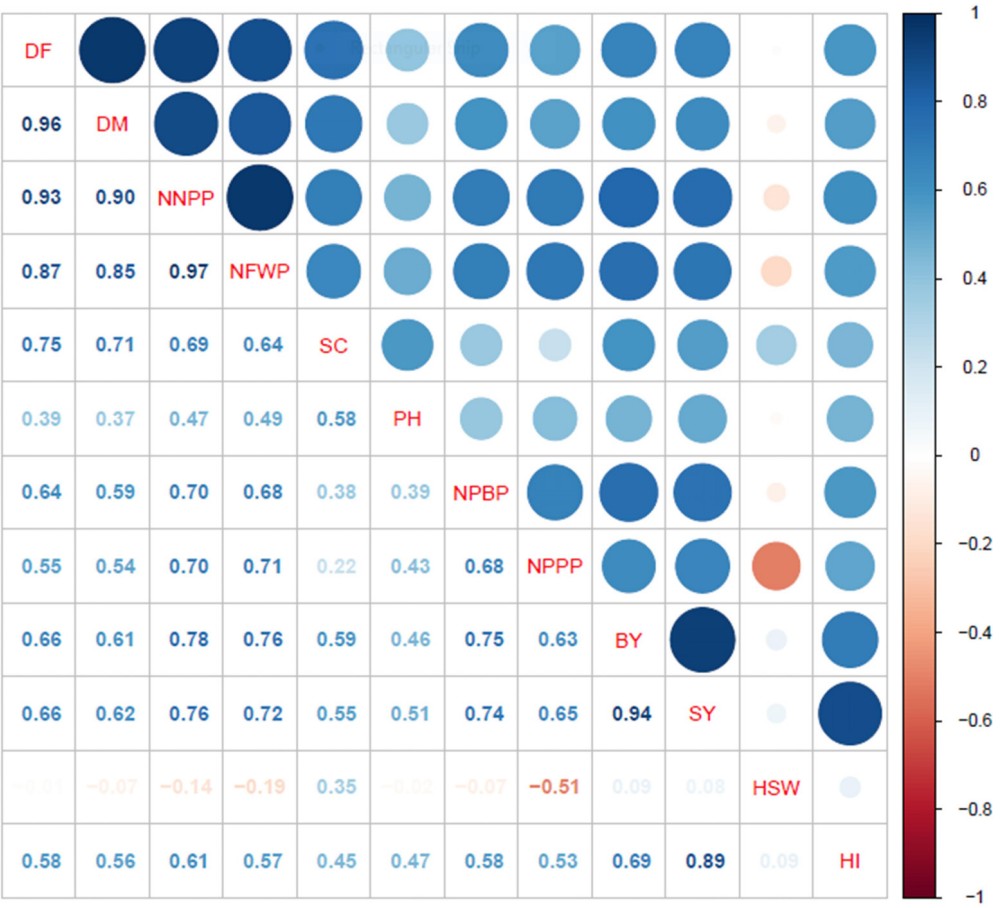

**Figure 3** **Correlation analyses for phenology, growth, yield components and yield attributes.** DF, days to 50% flowering; DM, days to 90% physiological maturity; NNPP, number of nodules per plant; NFWP, nodule fresh weight per plant; SC, stand count; PH, plant height; NPBP, number of primary branches per plant; NPPP, number of pods per plant; BY, biological yield; HSW, hundred seed weight; SY, seed yield; and HI, harvest index.

number of branch plant$^{-1}$. A similar line of work was also found by *Aklilu & Abebe (2020)*, who reported that years showed a highly significant effect in plant height, number of pods plant$^{-1}$, number of branches plant$^{-1}$, and seed yield for the three mungbean varieties.

## Implications of varieties and inoculation on phenological and growth components

### Implications on days to 50% flowering and 90% physiological maturity

Mungbean varieties behaved differently for days to flowering and maturity for non-inoculated and inoculated conditions. For instance, the Shewarobit Variety exhibited the same days to 50% flowering at both non-inoculated and inoculated conditions. However, the flowering date of inoculated NVL-1, Rasa, Boreda, and Arkebe varieties were significantly delayed by 4.01, 4.76, 5.40, and 7.21%, respectively, as compared to their non-inoculated counterparts. Additionally, varietal differences on days to flowering were observed in both non-inoculated and inoculated conditions. Consistently, Shewarobit

took 14.89, 14.89, 19.15, and 36.17% significantly longer days in the non-inoculated conditions and 11.34, 10.64, 14.53, and 31.21% longer days in the inoculated conditions for 50% flowering as compared to varieties NVL-1, Rasa, Boreda, and Arkebe, respectively. Similar to flowering, the inoculated NVL-1, Rasa, and Boreda varieties took longer days to physiological maturity as compared to their non-inoculated counterparts. In comparison to the varieties NVL-1, Rasa, Boreda, and Arkebe, Shewarobit's maturity was delayed by 11.50, 11.33, 10.83, and 21.67 days in the non-inoculated conditions and 7.67, 7.17, 7.83, and 18.83 days in the inoculated conditions, respectively.

The results of this study, generally, indicated the Rhizobium non-inoculated NVL-1, Rasa, Boreda and Arkebe varieties flowered and matured significantly earlier than their inoculated counterparts. The reason why longer days was taken in the inoculated NVL-1, Rasa, Boreda and Arkebe varieties compared to their non-inoculated counterparts might be due to the fact that Rhizobium bacteria inoculation of leguminous plants improves nitrogen content in the soil and plant tissues and promotes vegetative growth and thereby delaying flowering and maturity. However, non-significant effect was observed both on days to flowering and physiological maturity of Shewarobit. In both non-inoculated and inoculated conditions, Shewarobit was significantly flowered and matured later than the other varieties. This might be attributed to the fact that days to flowering and maturity of mungbean are controlled by Rhizobium inoculation and genetic characteristics of the crop. Previous studies showed that, the differential response to flowering and maturity of mungbean among varieties to different inoculation level was different. Correspondingly, *Ullah et al. (2016)* revealed significant differences among mungbean varieties for germination date under Rhizobium inoculation conditions. *Habte (2018)*, *Geletu & Mekonnen (2018)*, *Belay et al. (2019)*, *Aklilu & Abebe (2020)*, *Karaman & Kaya (2020)*, and *Dinsa et al. (2022)* also reported the varietal differences for days to flowering and maturity.

### Implications on number of nodules per plant

The study showed that varieties responses for the number of nodules plant$^{-1}$ were different for the different growing years. Remarkably, a higher number of nodules plant$^{-1}$ was recorded in the 2021 experiment than in 2022 for all varieties. Specifically, varieties Shewarobit, NVL-1, Rasa, Boreda, and Arkebe in the 2021 experiment provided 13.30%, 33.78%, 22.01%, 22.00%, and 24.09%, respectively, an increment of nodules plant$^{-1}$ than the 2022 experiment. Notable varietal differences for the number of nodules plant$^{-1}$ were also observed both in 2021 and 2021. For instance, in both years, varieties Shewarobit, NVL-1, Rasa, and Boreda showed comparatively good performance in bearing a higher number of nodules plant$^{-1}$ than the Arkebe variety.

Additionally, Rhizobium inoculation effect for the number of nodules plant$^{-1}$ for 2021 and 2022 was different, indicating the presence of interaction. A significant difference between inoculated and non-inoculated seeds was observed in the 2021-year experiment, with the inoculated seeds producing 4.03% higher numbers of nodules plant$^{-1}$ than their non-inoculated counterparts. However, no significant difference between inoculated and non-inoculated seeds was detected in the 2022-year experiment. Conspicuously, the non-inoculated seeds from the 2021 experiment produced 19.86% greater nodules plant$^{-1}$

than their non-inoculated counterparts in 2022, and the inoculated seeds in the 2021-year experiment brought 24.64% higher number of nodules plant$^{-1}$ than the inoculated seeds in 2022. The reason why the non-effectiveness of Rhizobium inoculation in 2022 might be due to the fact that the presence of unfavourable environmental conditions mainly high moisture deficiency for Rhizobium bacteria inoculation, nodule formation and growth. In line with this result, *Hungria & Vargas (2000)*, *Giller et al. (2013)*, and *Franke et al. (2018)* reported that high temperature and moisture deficiency causes nodulation failure in leguminous plants.

Mungbean varieties behaved differently for the Rhizobium bacteria inoculation levels. For example, the inoculated seeds of NVL-1, Rasa, and Boreda varieties significantly produced 4.76, 8.75, and 14.49%, respectively, higher number of nodules plant$^{-1}$ than the non-inoculated seeds. However, non-significant differences between inoculated and non-inoculated seeds were observed for the Shewarobit and Arkebe varieties. Overall, the Shewarobit variety at both inoculated and non-inoculated conditions showed a better nodule-bearing performance than the other four varieties. Specifically, at non-inoculated conditions, variety Shewarobit had 25.28, 38.98, 48.57, and 68.00% more number of nodules than varieties NVL-1, Rasa, Boreda, and Arkebe, respectively. Similarly, the former variety at inoculated conditions gave 20.95, 32.62, 39.40, and 65.24% higher numbers of nodules plant$^{-1}$ than the later varieties. The reason behind for the non-significant effect of Rhizobium bacteria inoculation on number of nodules plant$^{-1}$ for the Shewarobit variety might be due to the negligence or no reactions of genes/alleles of variety to the commercially produced Rhizobium inoculum. Similarly, *Musiyiwa, Mpepereki & Giller (2005)* reported that soybean varieties widely differed in their nodulation efficiencies with rhizobial strains.

### Implications on fresh-weight of nodules per plant

Similar to number of nodules plant$^{-1}$, mungbean varieties responded differently on fresh-weight of nodules plant$^{-1}$ for the different growing years. In 2021, significant differences on fresh weight of nodules plant$^{-1}$ were observed among all varieties. In particular, the Shewarobit variety recorded 11.48, 34.43, 42.62, and 59.02% higher fresh weight of nodules plant$^{-1}$ than varieties NVL-1, Rasa, Boreda, and Arkebe, respectively. However, in 2022, non-significant differences were observed between NVL-1 and Rasa varieties, while other variety pairs were significant. The Shewarobit variety in the 2022-year experiment gave 33.33, 40.74, and 50.00% higher fresh weight nodules plant$^{-1}$ than varieties NVL-1, Rasa, Boreda, and Arkebe, respectively. Additionally, all varieties gave higher fresh weight of nodules plant$^{-1}$ in the 2021 growing season than 2022.

Moreover, the interaction effect of Rhizobium inoculation and growing years was significant. Remarkable differences were found between Rhizobium inoculated and non-inoculated conditions for the 2021-year experiment. Accordingly, the inoculated mungbean seeds gave 13.04% higher fresh-weight of nodules plant$^{-1}$ than the non-inoculated ones. However, in 2022, no significant variation was observed between the inoculated and non-inoculated mungbean seeds. Both the non-inoculated and inoculated mungbean seeds of 2021-year experiment gave higher fresh-weight of nodules plant$^{-1}$ than the 2022-year experiment. The reason why higher fresh weight of nodules plant$^{-1}$ was recorded in

2021-year experiment than 2022 might be due to the favourable environmental conditions of 2021 mainly soil moisture and temperature for nodule-bearing and development in the 2022-year experiment. Corresponding report was forwarded by *Musiyiwa, Mpepereki & Giller (2005)* on soybean varieties.

### Implications on stand count

The main effect of varieties showed significant differences on stand count at physiological maturity. Notable significant differences on the survival rates between Arkebe and the other four varieties were observed. The NVL-1, Shewarobit, Rasa, and Boreda varieties produced 38.11, 36.97, 34.64 and 33.14%, respectively, a higher stand count than the Arkebe variety. Compared with other varieties, the death rate of Arkebe variety in the field was very high since after the completion of vegetative growth. Hence, the reason as why the lowest stand count was observed in Arkebe variety might be due to low resistance to competition effects and poor vegetative growth of the variety as compared to the others. This finding contrasts with the results reported by *Hussen, Asrat & Menzer (2020)*, who reported no significant variations of mungbean varieties for stand count at emergence and harvest.

### Implications on plant height

This study also showed that significant variation on plant height was observed between the Arkebe and Shewarobit and NVL-1 varieties. The tallest Shewarobit variety exceeded the shortest Arkebe variety by 3.61 cm. However, no significant difference in plant height was found among Shewarobit, NVL-1, Rasa, and Boreda varieties. The variation in height might be due to genetic characteristics of the varieties. This outcome is consistent with the report of *Kassa et al. (2022)*, who revealed that there was a significant variation in plant height among the genotypes of mungbean. Analogous output for plant height among mungbean varieties was also produced by *Belay et al. (2019)*, *Aklilu & Abebe (2020)*, and *Dinsa et al. (2022)*. In contrast to this result, *Nadeem, Ahmad & Ahmad (2004)* and *Hossain & Solaiman (2004)* reported plant height was affected significantly by Rhizobium inoculation. The occurrence of such contradictory reports might be due to the differences in genetic potential of the varieties, climatic and fertility conditions.

### Implications on number of primary branches per plant

In this study, it was generally observed that the effects of varieties of mungbean on different Rhizobium inoculation levels for the number of primary branches plant$^{-1}$ behaved differently. For instance, Rhizobium inoculation improved the number of primary branches plant$^{-1}$ for Rasa, Boreda, and Arkebe varieties. Thus, the inoculated seeds of these varieties yielded 13.92, 18.52, and 15.50%, respectively greater numbers of primary branches plant$^{-1}$ than their respective non-inoculated counterparts. However, non-significant differences were found between inoculated and non-inoculated seeds of the Shewarobit variety. In both inoculated and non-inoculated conditions, the Shewarobit variety consistently provided the highest number of primary branches plant$^{-1}$ while the Arkebe variety gave consistently the lowest. Under the non-inoculated conditions, variety Shewarobit produced 8.06, 15.13, 22.21, and 22.91% higher numbers of primary branches plant$^{-1}$ than the NVL-1, Rasa, Boreda, and Arkebe varieties, respectively. In the inoculated conditions, the differences

were less noticeable for the Shewarobit variety, exhibiting 0.72, 0.14, 3.30, and 7.59% higher numbers of primary branches plant$^{-1}$ than the other four varieties. Accordingly, *Aklilu & Abebe (2020)* reported significant mungbean varietal differences in the number of branches plant$^{-1}$.

## Implications of varieties and rhizobium inoculation on yield components and yield

### Implications on number of pods per plant

The study revealed that mungbean varieties' response for number of pods plant$^{-1}$ were different in different growing seasons/years. In 2021, significant variations were observed in all of the varieties' pairs tested, except between the Rasa and Arkebe varieties. Shewarobit in this year brought 24.17, 29.54, 30.11, and 38.86% higher number of pods plant$^{-1}$ than varieties NVL-1, Arkebe, Rasa, and Boreda, respectively. However, in 2022, non-significant differences were recorded between NVL-1 and Arkebe, as well as between Rasa and Boreda varieties. Similar to the 2021-year experiment, Shewarobit in 2022 produced 12.64, 19.80, 25.89, and 31.23% significantly greater numbers of pods plant$^{-1}$ as compared to varieties NVL-1, Arkebe, Rasa, and Boreda, respectively. Moreover, all varieties (Shewarobit, NVL-1, Rasa, Boreda, and Arkebe) in 2021 produced (24.97, 13.56, 20.44, 15.61, and 14.60%, respectively) significantly greater number of pods plant$^{-1}$ than in 2022.

The experiment also showed that varietal response was different in Rhizobium non-inoculated and inoculated conditions. The inoculated seeds of Rasa and Boreda varieties exhibited 14.37 and 16.12%, respectively, a higher number of pods plant$^{-1}$ than the non-inoculated counterparts. However, no significant variations were recorded between non-inoculated and inoculated seeds of the NVL-1, Shewarobit, and Arkebe varieties. Under non-inoculated conditions, Shewarobit revealed 21.68, 25.46, 36.17, and 43.29% greater number of pods plant$^{-1}$ than varieties NVL-1, Arkebe, Rasa, and Boreda, respectively. Conversely, no significant variations were found between non-inoculated seeds of NVL-1 and Arkebe, as well as Rasa and Boreda varieties. Similar to the non-inoculated conditions, the inoculated seeds of the Shewarobit variety produced 16.72, 19.91, 23.70, and 27.35% higher numbers of pods plant$^{-1}$ than varieties NVL-1, Rasa, Arkebe, and Boreda, respectively. Non-significant variations, however, were observed among the inoculated seeds of the NVL-1, Rasa, and Arkebe varieties.

Generally, both in non-inoculated and inoculated conditions as well as in 2021 and 2022 growing years, the Shewarobit variety provided the highest number of pods plant$^{-1}$ followed by NVL-1 and Arkebe varieties while the Boreda and Rasa varieties gave the lowest. The reason why the lowest number of pods plant$^{-1}$ were obtained from Boreda and Rasa over others might be due to the lowest branch-bearing nature of these varieties over others. Several studies also confirmed that mungbean genotypes had a significant effect on the number of plant$^{-1}$ (*Rasul et al., 2012*; *Ahmad et al., 2015*; *Wedajo, 2015*; *Habte, 2018*; *Aklilu & Abebe, 2020*; *Hussen, Asrat & Menzer, 2020*; *Baza, Shanka & Bibiso, 2022*; *Kassa et al., 2022*; *Dinsa et al., 2022*).

### Implications on biological yield

Similar to number of pods plant$^{-1}$, the response of mungbean varieties on biological yield in 2021-year experiment was different in the non-inoculated and inoculated conditions, indicating the presence of interaction. Significant variations between inoculated and non-inoculated conditions were observed for varieties Rasa, Boreda, and Arkebe. Thus, the inoculation of Rasa, Boreda, and Arkebe varieties increased the biological yield by 13.56, 19.98, and 20%, respectively. However, non-significant differences between inoculated and non-inoculated conditions were found for the Shewarobit and NVL-1. The reason for the non-significant differences between non-inoculated and inoculated seeds of the Shewarobit and NVL-1 varieties might be due to the negligence reactions of genes for inoculation in these varieties. Correspondingly, *Musiyiwa, Mpepereki & Giller (2005)* revealed that the soybean varieties performance differences with a rhizobial strains inoculations.

Though there was no significant variation with the Shewarobit variety, the non-inoculated NVL-1 variety provided 24.49, 27.37, and 49.09% higher biological yields than the non-inoculated Rasa, Boreda, and Arkebe varieties, respectively. Similarly, the inoculated NVL-1 variety exhibited 12.76, 16.51, 13.24, and 39.17% biological yield increments as compared to the inoculated Shewarobit, Rasa, Boreda, and Arkebe varieties, respectively. Overall, varieties NVL-1 and Shewarobit gave highest biological yields compared to the other varieties in both non-inoculated and inoculated conditions. While variety Arkebe gave the lowest biological yield. The decrease in biological yield of variety Arkebe due to poor vegetative growth, lowest hundred seed weight and shortest plant height might not have been compensated by the increase in other parameters such as number of primary branches plant$^{-1}$ and number of pods plant$^{-1}$. This might be the reason for the occurrence of the lowest biological yield on Arkebe varieties among others. Confirming this result, *Delić et al. (2011)* reported seed inoculation produced significantly higher grain yield (11–59%) and shoot dry weight (13–48%) in respect to untreated control. *Fernandes & Bhalerao (2015)* also proved that mungbean with Azotobacter species inoculation showed good morphological and biochemical attributes.

However, in 2022, only the main effect of variety showed significant effects on biological yield of mungbean. In this year, the Shewarobit variety produced 15.28, 15.39, and 24.88% significantly larger biological yields compared to Rasa, Boreda, and Arkebe varieties, respectively. Similar to the 2021-year experiment, the Shewarobit and NVL-1 varieties gave the highest biological yield in 2022 while lowest was recorded for Arkebe Variety. This result, generally, indicated that Rhizobium bacteria inoculation efficiency depend the genetic characteristics of the crop and environmental condition of the growing years. Equivalent findings were obtained by *Lema, Mekonnen & Gudero (2018)*, *Baza, Shanka & Bibiso (2022)*, and *Kassa et al. (2022)*, who verified that mungbean varieties (Gofa local, MH-97-6, and Sunaina) significantly differed for the above-ground dry biomass.

### Implications on hundred seed weight

Significant variations among varieties on hundred seed weight were also observed. Boreda, Rasa and NVL-1 varieties gave the highest hundred seed weights followed by the Arkebe variety, while lowest hundred seed weight was recorded for the Shewarobit variety. The

Boreda variety had 25.30 and 30.43% larger hundred seed weights than the Arkebe and Shewarobit varieties, respectively. Similarly, the NVL-1 variety produced 23.01 and 28.31% higher hundred seed weight than varieties Arkebe and Shewarobit, respectively. The Rasa variety also gave (23.94 and 29.18%, respectively) higher hundred seed weight than varieties Arkebe and Shewarobit. The Arkebe variety, in turn, yielded 6.88% larger hundred seed weight than the variety Shewarobit. However, no significant variations were found between Boreda, Rasa, and NVL-1 varieties for the hundred seed weights. This result might have been caused by the genetic characteristics of the crop since hundred seed weight is mostly internally controlled trait. Consequently, *Kaysha, Shanka & Bibiso (2020)* reported the Rasa variety gave a higher thousand seed weight (56.95 g) than that of the Shewarobit variety (45.32 g). Moreover, varietal differences in the hundred seed weight of mungbean were exhibited by several findings (*Belay et al., 2019*; *Aklilu & Abebe, 2020*; *Dinsa et al., 2022*; *Baza, Shanka & Bibiso, 2022*; *Kassa et al., 2022*).

### Implications on seed yield

Likewise to biological yield, varieties behaved differently in non-inoculated and inoculated condition in the 2021-year experiment. For instance, inoculated Rasa and Boreda varieties gave 19.10 and 23.13% larger seed yields than their respective non-inoculated counterparts. However, non-significant variation was found between inoculated and non-inoculated seeds of the Shewarobit and Arkebe varieties, although the inoculated Arkebe variety gave 24.28% higher seed yield than the non-inoculated ones. In the non-inoculated conditions, Shewarobit displayed significantly 27.08, 29.97, and 62.17% larger seed yields than Rasa, Boreda, and Arkebe varieties, respectively. Non-significant variation, however, was noted between varieties Rasa and Boreda. On the other hand, under inoculated conditions, no significant differences were observed among varieties Shewarobit, NVL-1, Rasa, and Boreda, which gave significantly higher seed yield than the Arkebe variety. *Sapna (2021)* reported Rhizobia isolate MR 63 and MB 17a sustained higher yields over other Rhizobia strains under both optimum and drought moisture regimes. *Dinsa et al. (2022)* also reported a significant interaction effect on varieties by year for seed yield and other yield components of mungbean.

Moreover, seed yield of mungbean was also affected by the interaction effect of varieties and growing years. The Shewarobit variety exhibited significantly 19.45, 21.02, 17.28, and 44.34% larger seed yields than varieties NVL-1, Rasa, Boreda, and Arkebe, respectively. No significant differences were found among NVL-1, Rasa, and Boreda varieties; however, these three varieties gave significantly 30.90, 29.53, and 32.71%, respectively, higher seed yield than variety Arkebe. The reason why the highest seed yield was recorded for variety Shewarobit over others, in both non-inoculated and inoculated conditions as well as 2021 and 2022 growing years might be due to the high branching habit and high pods-bearing natures of the variety over the others. A similar finding was obtained by *Aklilu & Abebe (2020)*, who reported the highest seed yield for the NVL-1 variety, followed by Shewarobit. The lowest seed yield was recorded for variety N-26 (Rasa). Similarly, *Habte (2018)*, *Aklilu & Abebe (2020)*, *Baza, Shanka & Bibiso (2022)*, *Dinsa et al. (2022)*, and *Kassa et al. (2022)*

revealed that there was a highly significant variation in seed yield among the genotypes of mungbean.

### Implications on harvest index

Shewarobit provided a significantly higher harvest index (14.70 and 25.07%) than the NVL-1 and Arkebe varieties, respectively. Rasa and Boreda had similar level of significance with Shewarobit; however, these varieties gave significantly 22.34 and 23.07%, respectively, higher harvest index than Arkebe. This outcome revealed that Shewarobit, Rasa and Boreda had the highest ability to convert the dry matter into economic yield compared to other varieties. Consequently, *Baza, Shanka & Bibiso (2022)* reported the notable differences of mungbean varieties on the harvest index.

## CONCLUSION AND RECOMMENDATIONS

The findings highlight that Rhizobium inoculation did not consistently enhanced the yield and yield components of mungbean across the two years of cultivation or among different varieties. To promote sustainable production and enhance mungbean productivity, commercial producers of Rhizobium bacteria inoculants and researchers should focus on evaluating the adaptability of both commercially produced and native inoculants across various growing environments and mungbean genotypes, ultimately leading to conclusive recommendations.

Overall, this study indicated that the highest seed yield across both growing years and varying levels of Rhizobium inoculation were achieved with the Shewarobit (660.90–1,214.74 kg ha$^{-1}$) and NVL-1 (532.35–1,097.78 kg ha$^{-1}$) varieties, followed by Rasa (521.99–1,095.00 kg ha$^{-1}$) and Boreda (546.70–1,106.63 kg ha$^{-1}$). The Arkebe variety (367.85–606.88 kg ha$^{-1}$), however, produced the lowest seed yield. Therefore, it can be concluded that the Shewarobit and NVL-1 varieties, along with Rasa and Boreda when inoculated with Rhizobium, demonstrated strong overall agronomic performance and are recommended for cultivation in the study area.

### Funding

This study received support from the University of Gondar. The funders had no role in study design, data collection and analysis, decision to publish, or preparation of the manuscript.

### Grant Disclosures

The following grant information was disclosed by the authors:
University of Gondar.

### Competing Interests

The authors declare there are no competing interests.

## Author Contributions

- Shiferaw Mebrat Delie conceived and designed the experiments, performed the experiments, analyzed the data, prepared figures and/or tables, authored or reviewed drafts of the article, and approved the final draft.
- Teferi Alem Adamu conceived and designed the experiments, performed the experiments, analyzed the data, prepared figures and/or tables, authored or reviewed drafts of the article, and approved the final draft.
- Abebaw Mulugeta Andualem conceived and designed the experiments, performed the experiments, analyzed the data, prepared figures and/or tables, authored or reviewed drafts of the article, and approved the final draft.

## Data Availability

The raw data is available in the Supplemental File.

## Supplemental Information

Supplemental information for this article can be found online at http://dx.doi.org/10.7717/peerj.19558#supplemental-information.

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
