# Peer review of "Adaptability and agronomic performance evaluation of mungbean (Vigna radiata (L.) Wilczek) varieties under non-inoculated and inoculated rhizobium bacteria conditions"

_PeerJ, doi:10.7717/peerj.19558_

## Round 0.1 · original submission · Major Revisions

The study is sound; however, the authors need to revise the manuscript carefully in light of the reviewer's comments.

Reviewer 1 ·

Basic reporting

The study is interesting and worth publishing. Manuscript is well written with professional English used.
Background is enough but it should focus more on the topic of the study rather than generalized information.
Results are well written, however, discussion needs incorporation of mechanism which improved the growth and yield parameters of the mung bean.

Experimental design

Experimental design is well constructed with respect to field experiment.

Validity of the findings

As the reported varieties’ agronomic and yield performances have not yet been assessed and evaluated in the Central Gondar Zone, therefore, it is a new study in the respective area of the study.

Additional comments

Reviewers’ comments
Introduction should be focused on the topic of the study. Irrelevant material should be removed
1. In introduction, line 89, “N” should be in full as “nitrogen (N)”.
2. Line 149-155 make no sense, these sentences should be removed.
3. Line 196, 198, “x” should be replaced with proper symbol “×”.
4. Line 196, 198, make corrections in “m2 as “m2”.
5. Line 274, 275, make corrections in plant-1 as “plant-1”
6. Discussion section need revision by providing reasons to elaborate the mechanism that how treatments affect positively to improve growth and yield attributes of mung bean.
7. overall manuscript should be checked for grammatical and typo mistakes.

Reviewer 2 ·

Basic reporting

When evaluated in terms of scientific validity and suitability for inclusion in the academic literature, it is suitable for publication after minor corrections.

Necessary corrections have been made on the file.

Experimental design

Minor corrections in the trial setup and data collection are indicated in the file.

Validity of the findings

tables and graphs are adequately explained and minor spelling corrections are indicated in the additional file.

Additional comments

In the references section, there are literatures that are not in the text. there are also literatures that are not in the references section in the text and these are specified in the file.
The authors have not corrected the references section according to Peer J rules. this correction should be made.
Although the study is not a new and original subject worldwide, it will contribute to the literature since it was carried out on the region and new varieties.

best regards

Annotated reviews are not available for download in order to protect the identity of reviewers who chose to remain anonymous.

·

Basic reporting

1. The research idea is strong, and the availability of data for two growing seasons is a key strength of the study. Please include the major reason for comparing data across the two seasons, such as accounting for seasonal variability or validating the consistency of results. This would further enhance the clarity and impact of the study

2. The title requires rearrangement for better clarity and alignment with the study's focus. Additionally, the writing should be carefully reviewed to ensure precision and adherence to the journal's guidelines. Please address these aspects thoroughly before submission to meet the journal's standards.

3. Please include line numbers in the manuscript to facilitate easier referencing and review of specific sections

4. In the abstract, could you provide a numerical comparison of the significant data (e.g., means, percentages, or effect sizes) for the interactions and main effects reported in the results, particularly for days to flowering, maturity, yield components, and inoculation effects across the two years?

Experimental design

1. The experimental design is well-structured, but it could be further improved by illustrating the treatments and design in a pictorial format for better clarity. Additionally, representing the design as a 5 × 2 factorial RCBD in the context would enhance understanding and alignment with standard experimental design terminology.

2. The major characteristics of the mungbean varieties (e.g., growth habit, maturity period, yield potential, stress tolerance) are not described. Including these details with relevant references would enhance the manuscript and clarify how varietal differences may impact the results

Validity of the findings

1. Could you confirm whether the nodule count is indeed below the expected average? Or if not please confirm this.

2. Could you clarify which specific environmental factors (e.g., temperature, rainfall) were unfavorable in 2022 compared to 2021, and how they might have caused the reduced growth and yield parameters observed?

Additional comments

1. Uninoculated can be replaced with non-inoculated in all the manuscript
2. In the Problem statement and justification of Introduction, it would be better to avoid using too many small paragraphs. Instead, structure the content into 3–4 or even less well-explained paragraphs.
3. The description of the study area is comprehensive but exceeds the required level of detail, making it difficult for readers to focus on the most relevant information. Organize Chronologically or Thematically.
4. Why was sowing not conducted in the same week for both years (2021 and 2022) during the two-season cultivation of mungbean? Was this variation in sowing dates.
5. The content appears well-written, but I am concerned whether using 1000 seed weight is appropriate for mungbean.

---

## Round 0.2 · accepted · Accept

After careful review, it is our pleasure to inform you that your manuscript has been accepted for publication. All comments and concerns raised during the review process have been satisfactorily addressed by the author, and the revised manuscript meets the publication standards.

Congratulations on your work!

Reviewer 1 ·

Basic reporting

The author has revised the manuscript carefully by addressing all the queries and have incorporated all suggested corrections in the manuscript. Literature is sufficient including figures and tables.

Experimental design

Experiment is well designed addressing the relevant research question. Methodology is sufficient with all necessary detail.

Validity of the findings

Data providing the information about the study is sufficient and statistically sound. The discussion section has been improve by addition relevant literature and conclusion is well written.

Additional comments

The manuscript has been revised carefully according to the suggestions of the reviewers, I would recommend its acceptance.

Reviewer 2 ·

Basic reporting

Clear and unambiguous, professional English used throughout.

Literature references, sufficient field background/context provided.

Professional article structure, figures, tables. Raw data shared.

Experimental design

Original primary research within Aims and Scope of the journal.
Research question well defined, relevant & meaningful. It is stated how research fills an identified knowledge gap.
Rigorous investigation performed to a high technical & ethical standard

Validity of the findings

Impact and novelty not assessed. Meaningful replication encouraged where rationale & benefit to literature is clearly stated.
All underlying data have been provided; they are robust, statistically sound, & controlled.
Conclusions are well stated, linked to original research question & limited to supporting results

Additional comments

the work can be published as it is.